# A Proposed Methodology for a Risk Assessment-Based Liposome Development Process

**DOI:** 10.3390/pharmaceutics12121164

**Published:** 2020-11-29

**Authors:** Zsófia Németh, Edina Pallagi, Dorina Gabriella Dobó, Ildikó Csóka

**Affiliations:** Faculty of Pharmacy, Institute of Pharmaceutical Technology and Regulatory Affairs, University of Szeged, Eötvös u. 6., H-6720 Szeged, Hungary; zsofia.nemeth@pharm.u-szeged.hu (Z.N.); edina.pallagi@pharm.u-szeged.hu (E.P.); or dobo.dorina.gabriella@szte.hu (D.G.D.)

**Keywords:** quality by design, quality planning, risk assessment, critical factors, liposome formulation, thin-film hydration method

## Abstract

The requirements of a liposomal formulation vary depending on the pharmaceutical indication, the target patient population, and the corresponding route of administration. Different preparation methods require various material attributes (MAs) (properties and characteristics of the components) and process parameters (PPs) (settings of the preparation method). The identification of the quality target product profile for a liposome-based formulation, the critical quality attributes of the liposomes, and the possible MAs and PPs that may influence the key characteristics of the vesicles facilitates pharmaceutical research. Researchers can systematise their knowledge by using the quality by design (QbD) approach. The potential factors that influence the quality of the product can be collected and studied through a risk assessment process. In this paper, the requirements of a liposome formulation prepared via the thin-film hydration preparation technique are presented; furthermore, the possible factors that have an impact on the quality of the final product and have to be considered and specified during the development of a liposomal formulation are herein identified and collected. The understanding and the application of these elements of QbD in the pharmaceutical developments help to influence the quality, the achievements, and the success of the formulated product.

## 1. Introduction

Liposomes are described as artificially prepared vesicles composed of one or more concentric lipid bilayers that are enclosing one or more aqueous compartments by the European Medicine Agency [1]. Liposomes as drug carrier systems have several advantages [2]. These formulations can be used, among others, to protect active pharmaceutical agents (API), incorporate both lipophilic and hydrophilic drug molecules, and maintain targeted drug delivery [3]. From the beginning until the present day, four different generations of liposomes have been distinguished. The first-generation liposomes (conventional liposomes) are made up of neutral and/or negatively charged phospholipids and cholesterol [4]. These vesicles are taken up by the reticuloendothelial system (RES) (phagocytes) in cases of intravenous administration; thus, their circulation time is short [5]. The second generation consists of long-circulating liposomes, while the third generation is made from surface-modified liposomes that can avoid the defence mechanism of the immune system. The fourth generation is built up from polyethylene glycol (PEG)ylated or the so-called “stealth” liposomes [3,4]. The surface of these vesicles is coated with a hydrophilic polymer, such as polyethylene glycol (PEG), that increases the repulsive forces between the liposomes and thus avoids the protein adsorption and opsonisation of the liposomes by the RES [5,6]. In this way, longer residence time is provided for the liposomes to remain in the tumour tissues [6]. Beyond the generational grouping of the liposomes, they can be classified regarding their compositions and drug delivery mechanisms such as conventional liposomes, long-circulating liposomes, polymorphic or bioresponsive liposomes [7,8,9] (pH-sensitive, thermos-sensitive, cationic liposomes), and decorated liposomes (surface-modified vesicles and immunoliposomes) [10,11]. Liposomes are used for the application of highly potent medications. Their pharmaceutical application is essential in the field of cancer therapy, besides that of the already marketed liposomal drugs in this field, and several new studies are in progress in the above-mentioned and newly targeted medical areas as well [12,13,14]. Nano-system development, including nanoscale liposome research, is receiving increasing attention nowadays. Nano-sized liposomal formulations can play a highly focused role in the therapy development of unmet clinical needs and diagnostic imaging techniques in the future. However, the regulatory authorities need to meet several challenges in terms the quality, safety, and efficacy aspects of the liposome-based products [15,16]. There is still no well-defined regulatory authorisation process for liposomes; however, several international groups are working on this. The International Organisation for Standardisation (ISO) defined the nanoscale size as the range extending between 1 and 100 nm [17]. On the basis of their definitions, nanoparticles are those nano-objects that have all of their external diameters in the nanoscale, and there is no significant difference between the lengths of the longest and shortest axes of the particle [17]. Therefore, the size of the liposomes and their homogeneity (size distribution) are fundamental features of the systems. The polydispersity index (PdI), a dimensionless value theoretically between 0.0 and 1.0, provides information about the uniformity of the particles. PdI values less or equal to 0.3 are supposed to be the indicator of distribution with acceptably low polydispersity. In the case of lipid-based nanocarriers, formulations with a PdI of 0.3 and below are acceptable and are an indicator of a homogenous population of the vesicles [18]. The zeta potential value is used to define the repulsion or the attraction between the vesicles, and in this way, to predict the stability of the liposome system [19]. Liposomes with an average surface charge higher or equal to 10 mV in absolute value are considered as negative or positive vesicles, while between these values are considered neutral liposomes [20]. Nanoparticles with zeta potentials higher than +30 mV or lower than -30 mV are considered as a stable system [21]. The lamellar structure of the liposomes can also have an impact on their therapeutic application (e.g., incorporated API selection, dosage form selection, administration route definition).

The quality by design (QbD) approach is a quality management concept in the pharmaceutical industry that focuses on the prior definition and design of the target product considering all of the needs and requirements emerging from the clinical side (patient), the industrial processes, and the regulatory aspects [22,23]. QbD is a systemised, structured, knowledge- and risk assessment-focused approach, and the potentials of its extension have previously been shown by Csóka et al. [24]. The QbD approach is efficiently applicable during nano-pharmaceutical research as well [25,26,27,28,29,30]. The development process of the liposomes is challenging due to their complex manufacturing processes. The tools of the QbD approach can guide the formulation process to obtain higher quality liposomal products [31].

The whole QbD method is specified in the guidelines of the International Council for Harmonisation of Technical Requirements for Pharmaceuticals for Human Use (ICH) [32,33,34] Briefly, the QbD method includes the following general steps:

(1) Quality target product profile (QTPP) definition: the QTPP is a prospective summary of the quality characteristics of the drug product that ideally will be achieved to ensure the desired quality, taking into account the safety and the efficacy of the drug product, considering, e.g., the route of administration, the dosage form, bioavailability, strength, and stability [33]. 

(2) Identification of the critical elements, such as the critical quality attributes (CQAs) of the targeted product, critical material attributes (CMAs), and critical process parameters (CPPs), which are related to the selected production method. According to the definition of the ICH guideline, a CQA is a physical, chemical, biological, or microbiological property or characteristic that should be within an appropriate limit, range, or distribution to ensure the targeted product quality. CQAs are generally associated with the drug substance, the excipients, the intermediates (in-process materials), and the drug product [33]. A CPP is a process parameter that variability has an impact on the CQAs and therefore should be monitored or controlled to ensure that the process produces the targeted quality [33]. 

(3) Risk assessment (RA): RA is a valuable science-based process that is used to identify and rank the parameters on the basis of their impact on the CQAs of the product. Risk assessment is typically performed as the first step during an early phase of the pharmaceutical development processes and is evaluated again when more information becomes available and higher knowledge is obtained [32,33]. The current experimental knowledge obtained from the former practical studies have to be aligned with information from the relevant literature. To perform a successful RA, first, the research team has to define the precise target product (QTPP) and then has to select the critical factors and estimate the interdependence of the critical factors, ranking them by the severity of their impact. The team members estimate the level of the interactions between the parameters occurring during the formulation process (production settings, materials, etc.). All the elements applied in the RA (QTPP elements, CQAs, CMAs, and CPPs) are defined and selected by the research group; therefore, their knowledge strongly impacts this selection process. Risk is defined as the combination of the probability of the occurrence of harm and the severity of that damage. The RA is a systematic process to evaluate the necessary information for the support of the risk-defining step within the risk management process. It means the identification of hazards and the analysis of risks [31]. The quality risk management tools provide systemic and reproducible methods based on up-to-date knowledge to rate the probability, severity, and sometimes detectability of the risk. These methods can be qualitative or quantitative. Once the risk is expressed quantitatively, a numerical scale is assigned for evaluation [33]. The numeric score of the evaluated risks could arise from the multiplications of the severity and occurrence (or probability) values, or sometimes from the severity, occurrence, and detectability if the same scale was used for the estimation of all of these parameters. The RA software can help in this process, but even during the software-supported assessments, the identification of the risks and the estimations of the severity and the occurrence are the task and responsibility of the research group. The software only makes the calculations and provides the data assessment and visualisation of the final results. These results are the basis of the design of experiments (DoE). 

(4) Design space (DS) development: DS is a multidimensional combination and interaction of the input variables (e.g., material attributes) and the process parameters that have been demonstrated to assure quality. 

(5) Definition of the control strategy. 

(6) Life cycle management. 

For better understanding, the schematic structure of the QbD approach is presented in Figure 1.

This paper aimed to collect and evaluate the parameters that influence the manufacturing process of a liposomal pharmaceutical product in order to help the researchers and the professionals in the pharmaceutical industry in the QbD-based new liposome design and development. The authors aim to present a wide range of potential QTPP and CQA elements and their characteristics to highlight the potential decision and target points. It was also intended to give an example of how to use RA to rank the influencing parameters. For this illustration, the thin-film hydration method [35], the most common liposome production process (Table 1), was chosen, as the authors have practical experience and knowledge about this technique from their previous studies [27]. This method was described for the first time and used to prepare the first liposomes by Alec Douglas Bangham and his colleagues in 1965 [35]. Several modified versions of the original technique exist (Table 1), however, the basic steps of the process are mutual [36]: (1) preparation of the lipid film from phospholipids and cholesterol, (2) hydration of the thin film with a hydration medium, and (3) modification of the numbers of layers and the size of vesicles. 

## 2. Methods

The LeanQbD software (QbD Works LLC, Fremont, CA, USA) was used for the RA procedure. The first element of this procedure was the interdependence rating between the QTPPs and the CQAs, and the CQAs and the CPPs. A three-level scale was used to describe the relation between the parameters: “high” (H), “medium” (M), or “low” (L). In the software, the qualitative three-level scale, used for the estimation, is linked to a selectable numeric scale (0–10, or 0–100), which gives, at the end, the severity scores of the evaluated risk factors on the basis of mathematical calculations. In this study, the 0–10 scale was used. After the categorisation of the interdependence, a risk occurrence rating of the CPPs (or probability rating step) was made, applying the same three-grade scale (H/M/L) for the analysis. As the output of the initial RA evaluation, Pareto diagrams [37] were generated by the software, presenting the numeric data and the ranking of the CQAs and the CPPs according to their potential impact on the aimed final product (QTPP). The Pareto charts not only show the differences of the CMAs and the CPPs by their effect but also help to select the factors of a potential experimental design.

## 3. Results

Table 2 summarises the potential QTPP elements collected by the authors. Potential CQAs are collected and presented in Table 3.

As the preparation method (Table 1) defines the CPPs of the liposome formulation process, a production technique that provides the target CQAs need to be selected prior to the investigation of CMAs and CPPs. The API can be added to the formulation via passive or active loading techniques [3]. Mechanical dispersion [3,19,38,39], solvent dispersion [3,38,39], and detergent removal [3,38,39] methods belong to the passive loading techniques, in which methods the lipid films are prepared via different techniques, hydrated to obtain liposomes, and the drug is captured during the manufacturing process [3,39]. In case of active loading, the API is incorporated into the already prepared liposomes via gradient loading techniques using buffers or ammonium sulphate gradients [39]. Besides the conventional preparation methods, there are also numerous approaches that have been recently developed to produce liposomes [39,40]. In this paper, the thin-film hydration method-related factors are presented. The potential CMAs and CPPs of the technique are systemised in a flow chart in Figure 2. The steps of the thin-film hydration method [36] are shown in the middle of the figure, while the related material attributes (MAs) and process parameters (PPs) are presented on the two sides of the chart.

The general criticality of the presented factors was investigated in a RA, and the rankings of the elements of CQAs, illustrated with Pareto charts for better understanding, are shown in Figure 3, while CMAs and CPPs are shown in Figure 4.

## 4. Discussion

The QTPP (Table 2) depends mainly on the therapeutic/clinical aims and requirements, as well as the characteristics of the drug substance, and it is always unique. For instance, QTPP may be a nano-sized liposome-containing injection for cancer therapy with a proper dose of drug and drug release dedicated to the therapeutic needs. Those quality attributes that are critically related to the QTPP are the CQAs. That is the reason why the CQAs are also always unique and depend on the QTPP. The potential CQAs (Table 3) are, e.g., the type of the liposome, its lamellar structure, vesicle size, size distribution, sterility, viscosity, and stability, or the dissolution profile of the formulation. The API encapsulation efficiency is also a critical attribute for the liposomes, in addition to the zeta potential, which refers to the stability of the vesicles. PdI is one further potential CQA for lipid-based nanocarrier systems such as liposomes.

The application of a quality management visualisation tool, such as a fishbone diagram, process mapping, or a flow chart, is always useful for the identification of the CMAs and the CPPs of the aimed liposomal product. In this case, to show the systemic collection and presentation of the potential CMAs and CPPs, we built a flow chart (Figure 2). In the middle of the figure, the steps of the production process, which in this case was the thin-film hydration liposome preparation method, are presented. The left side of the flow chart contains the material attributes (MAs), and the right side shows the process parameters (PPs). These MAs and PPs can affect the result of the thin-film hydration-based liposome manufacturing process. The critical ones have to be selected and labelled as CMAs and CPPs. To make this figure and the tables of QTPP and CQAs, prior knowledge, previous experimental experience, and a thorough literature background survey of the field [31,41,42,43,44,45,46,47,48] were necessary. Although, the main points of the tables and figures are shreds of evidence from the literature mixed with practical experiences, the systemic collection of all the relevant factors and data in one paper is the novelty of the work. The demonstration of the CMAs and the CPPs parallelly enhances the transparency of their relationships. In the following step, RA can be performed among the elements of the QTPP, the CQAs, and the CMAs and the CPPs. Several tools are suitable for an RA, e.g., the support of an RA software can help to achieve proper and quick implementation. In the presented case, the LeanQbD (QbD Works LLC, Fremont, CA, USA) RA software was applied. The interdependence rating among the elements was made on a three-grade scale, as the interaction is low (L), medium (M), or high (H). This process was made step by step for each pair of factors on the basis of the prior experimental and literary knowledge. The results of the RA are presented in Pareto charts generated by the software (Figure 3 and Figure 4). Figure 3 shows the theoretical ranking of the CQAs of the liposomes according to the initial general RA made by the authors. It may also vary in other cases on the basis of the QTPP. Figure 4 presents the general ranking of the CMAs and the CPPs depending on their severity for the liposomal product. It may vary on the basis of the QTPP and the CQAs. According to the RA, the most influential CMAs, organised in descending order, are the phospholipids, the API content [27], the surface modifiers, the cholesterol content, the ratio between the phospholipids and the cholesterol, the phase transition temperature of the lipids, and the quality of the hydration media and the cryoprotectant, while the CPPs are the working temperature, the duration of the sonication, and the number of filtrations. The effect of the CMAs/CPPs can be accurately investigated if some of the values are set on the same level, while the ones under the scope of the study are changed according to the DoE.

Xu et al. performed a risk analysis study on liposomes prepared using the thin-film hydration technique and loaded with superoxide dismutase via a freeze–thaw cycling technique. They analysed those factors that affect the size, the encapsulation efficiency, and the stability of the liposomes. For this evaluation, they checked the properties of the formulation, the process, the analytical method, and the instrumentation reliability. They found that the “analytical method” and the “instrument reliability” categories can be well-controlled; therefore, the factors of these two categories are not critical. However, the factors of the “analytical method” and the “instrument reliability” are non-negligible for the selection and settings of the characterisation methods. Their findings, namely, the influencing role of the lipid concentration, the cholesterol ratio, and the quality of the phospholipids are consistent with our results [49]. Porfire et al. provided a general overview of the QbD approach for liposomes without defining a production process and described methodologies for liposome characterisation as a control strategy in detail. Their reasonable considerations were built into the tables of this paper with our additions. The facts above draw attention to the low number of studies following the steps of the QbD recommended by the regulatory authorities [31]. Our presented work fits well into this scientific research area; it extends the previous knowledge and gives a detailed overview of the QbD application. The systemised and structured form of the facts and information may help researchers in designing and planning their future studies of liposomes.

## 5. Conclusions

This work aimed to collect and systemise all the relevant factors of the liposome formulation development via the QbD technique. The application of the QbD approach is a regulatory requirement in the pharmaceutical submissions, and in these applications, RA is the key step. In this study, the theoretical method was presented, the potential QTPP elements of the liposome-based formulations were determined, and the potential CQAs of the liposomes were also collected. The potential critical material attributes and process parameters that need to be considered during the formulation design of the thin-film hydration liposome preparation method were listed and evaluated. The method of screening was also presented to identify the most critical factors. The phospholipids, the API content, the surface modifiers, the cholesterol content, the ratio between the phospholipids and the cholesterol, the phase transition temperature of the lipophilic phase, and the quality of the hydration media and the cryoprotectant were found to be the CMAs of highest influence. Furthermore, the working temperature, the duration of the sonication, and the number of filtrations were identified to be essential CPPs. The authors believe that the presented concept may help researchers to establish and perform studies on liposomes with less effort and more success.

## Figures and Tables

**Figure 1 pharmaceutics-12-01164-f001:**
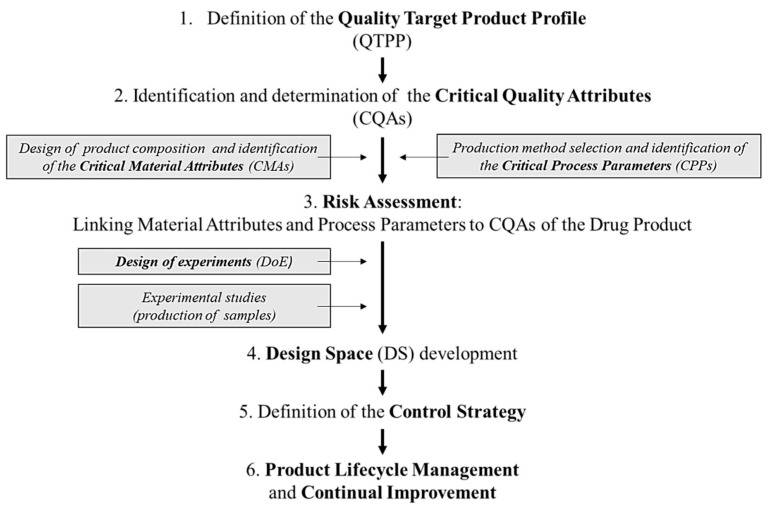
Schematic structure of the quality by design (QbD) approach.

**Figure 2 pharmaceutics-12-01164-f002:**
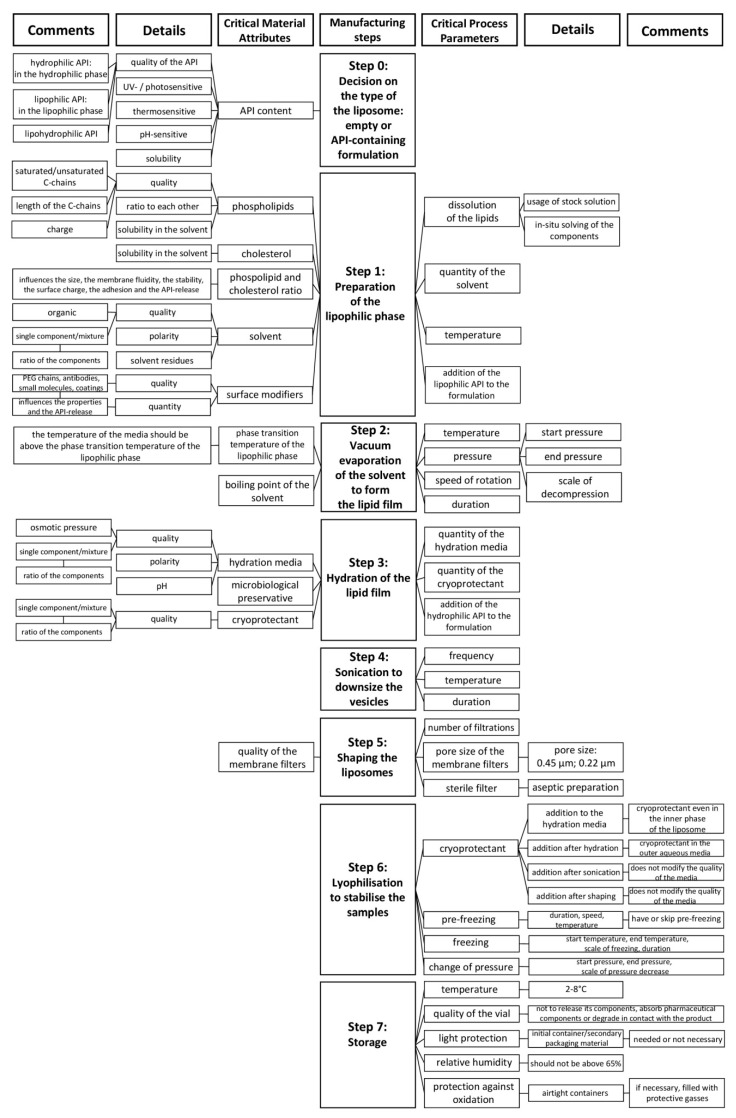
Collection of the properties of the liposome components (material attributes (MAs)) and the preparation method (process parameters (PPs)) that affect the result of the thin-film hydration liposome manufacturing technique.

**Figure 3 pharmaceutics-12-01164-f003:**
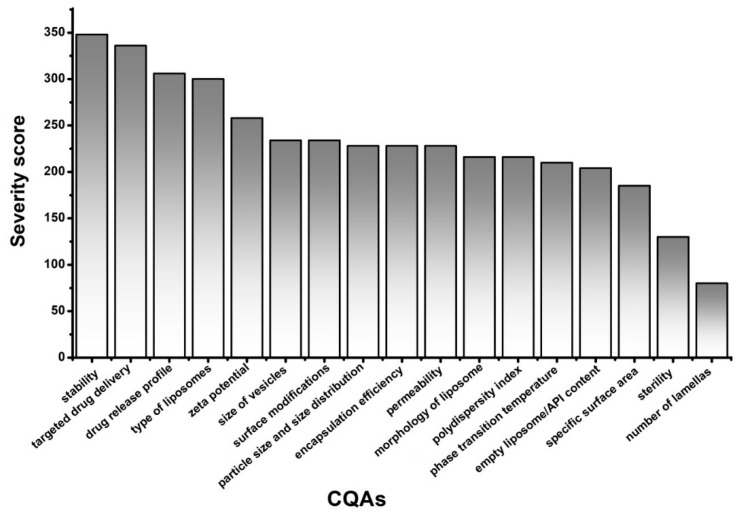
Rankings of the critical quality attributes (CQAs) of the liposomes.

**Figure 4 pharmaceutics-12-01164-f004:**
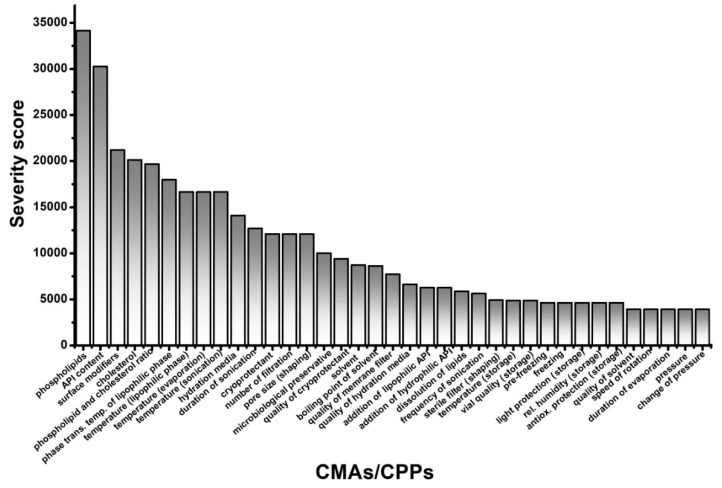
Rankings of the critical material attributes (CMAs) of the liposome components along with the critical process parameters (CPPs) of thin-film hydration.

**Table 1 pharmaceutics-12-01164-t001:** Potential methods to prepare liposomes.

Preparation Methods	Subtypes	Comments
Mechanical dispersion methods	probe or bath sonication	− the critical parameters vary on the basis of the selected preparation method; therefore, the definition of the production technique has to be the first step of every liposome formulation process − the properties of the liposomes (e.g., number of lamellas, size, and distribution of vesicles)
	French pressure cells—extrusion	
	freeze-thawed liposomes	
	membrane extrusion	
	lipid film hydration techniques	
	hydration of proliposomes	
	micro emulsification, coalescence of small vesicles	
	dual asymmetric centrifugation	
	heating method, Mozafari method	
	electro-formation	
Solvent dispersion methods	ether injection	
	ethanol injection	
	reverse-phase evaporation	
	solvent spherule method	
Detergent removal methods	dialysis	
	detergent removal of mixed micelles	
	gel-permeation chromatography	
Novel methods	microfluidisation	
	supercritical-assisted method	
	freeze-drying of double emulsions	
	membrane contractor method	
	curvature-tuning	
	biometric reaction for vesicular self-assembly	

**Table 2 pharmaceutics-12-01164-t002:** Collection of the possible factors of the quality target product profile (QTPP) for a liposome-based formulation.

QTPP Factors	Details	Comments
Indication/therapeutic effect	based on the API	its characteristics may necessitate the use of liposomes
	not important for empty liposomes	empty liposomes are used, e.g., in cosmetology
Target patient population	based on the indication	applicable for each age group in the suitable dosage form
Route of administration	the composition may differ on the basis of the target	can be determined by the API and the target patient population
Site of activity/target	based on the indication	targeted delivery
	based on the API	
Dosage strength	based on the API	differs even in the same pharmaceutical subgroup
	based on the target patient population	needed dose changes with age and health condition
	based on the indication	appear in the case of preparation with a wide range of indications
	based on the administration route	e.g., in the case of nasal application, the needed dose is less than per os
Dosage form/appearance	liposomes in aqueous solution	transparent, light scattering liquid (vesicles in colloid size)
	lyophilised powder	solid powder; colour based on the API and the excipients
Viscosity	based on the administration route	sign of stability; maintains efficient drug release; higher viscosity indicates a smaller size, a narrow PdI, slower drug release, and lower clearance rate
Osmolarity	based on the administration route	be tolerable, ideally 300 ± 30 mOsm/kg
Physical attributes of the liposomes	morphology, particle size, and zeta potential	change with the adjustment of the composition
Pharmacokinetics	liberation, adsorption, distribution, metabolism, elimination	necessary mostly for API-loaded liposomes
Safety	complement activation-related pseudoallergy (CARPA)	all types of intravenous liposomes can cause CARPA; enhanced by increasing size in the 70–300 nm range; more than 71 mol% cholesterol; PEG-PE insertion
	chemical/biological decomposition	needs to be investigated
	degradation products	concentration must be under the legal limit
Sterility	based on the administration route	sterile and pyrogen-free or aseptic preparation is not needed
Stability	in aqueous solution	needs to be stable; duration of stability is decisive
	in freeze-dried powder form	
Solubility/dissolution	in aqueous solution	media: non-toxic, non-irritable
	in freeze-dried powder	immediate release
Homogeneity	homogenous formulation	sign of stability
Drug release	based on the treatment	site and timing can be modified

**Table 3 pharmaceutics-12-01164-t003:** Collection of the possible factors of critical quality attributes (CQAs) of liposomes.

CQAs	Details	Comments
Type of liposomes	conventional liposomes	neutral or negative phospholipids
	immune liposomes	antibodies, antibody fragments
	cationic liposomes	positive phospholipids
	magnetic liposomes	metal particles
	bioresponsive liposomes	thermosensitive (37 °C < Tm)
		pH-sensitive (acidic milieu)
		LiPlasome (secretory phospholipase A2)
Number of lamellas	more layers	multilamellar (>0.5 µm)
		oligolamellar (0.1–1.0 µm)
	one layer	unilamellar
Size of vesicle	small unilamellar vesicle (SUV)	20–100 nm
	medium-sized unilamellar vesicle (MUV)	between SUV and LUV, >100 nm
	large unilamellar vesicle (LUV)	>100 nm
	giant unilamellar vesicle (GUV)	>1 µm
Surface modifications	no modification	rapid elimination
	polyethylene glycol (PEG) chains (stealth liposomes) (quality and quantity of the chains)	steric exclusion (decreased opsonisation and phagocytosis); prolonged circulation
	monoclonal antibodies, antibody fragments, peptides, nucleic acids, carbohydrates, small molecules	provide targeted delivery by biding to the targeted receptors
Morphology of liposomes	spherical vesicles	self-organised structure
	concentric layers	multi-layered vesicles
	spherical with multiple non-concentric lipid vesicles inside	multivesicular liposome (MVL)
Particle size and size distribution	d(0.1), d(0.5), d(0.9), span, surface weighted mean (D[3,2]), volume weighted mean (D[4,2])	mean particle size should be under 200 nm; ideal around 100 nm
Polydispersity index (PdI)	indicating polydispersity of the system	below 0.5 is acceptable
Specific surface area (SSA)	influences drug release	smaller vesicles maintain higher surface area-to-volume ratio than the larger particles
Zeta potential	indicating stability	stable formulation around ±30 mV
Phase transition temperature (T_m_)	influences drug release	determined by the composition of the liposome; cholesterols reduce the value
Empty liposomes/API content	modifies the physical attributes of the liposomes	the characteristics of the API determine its position
Position of the API	hydrophilic API	in the hydrophilic aqueous centre
	lipophilic API	in the lipophilic double membrane
	surface-bounded	monoclonal antibodies, antibody fragments, peptides, nucleic acids, carbohydrates, small molecules
Encapsulation efficiency (EE)	higher EE% is the goal to increase the drug concentration in the final formulation	manufacturing costs can be reduced, and more flexible dosing can be provided by higher EE%
Permeability	semi-permeable membrane	the highest permeability is at T_m_;
targeted drug delivery	target specificity	increases effectiveness
Drug release profile	maintains therapeutic activity	site and timing can be modified
Sterility	if necessary	even for the materials
		in the case of aseptic preparation
Stability	chemical, biological, microbiological	characteristic values must remain
		in the recommended ranges until use

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
