# Peer review of "A Proposed Methodology for a Risk Assessment-Based Liposome Development Process"

_pharmaceutics, 2020, doi:10.3390/pharmaceutics12121164_

Round 1
Reviewer 1 Report
In the paper “A Proposed Methodology for A Risk Assessment-Based Liposome Development Process” the authors describe an approach to systematically assess the requirement and critical parameters to optimize and tailor the liposomal formulation for a specific application. The quantification of the parameters impact is novel and interesting and is achieved using the Quality by Design (QbD) method whose components are clearly defined in the introduction. The chosen factors of product profile and quality attribute are well listed and justified. The results are also shown and discussed in a clear way and seem legit, even if not so novel. Anyway, it is not clear how the Risk Assessment (RA) is performed including within the software the previous knowledge, previous experimental experiences and literature background. Moreover, is not specified how the quantification and relative relevance, that represent the main novelty of the paper, are quantified and how the described method is able to quantify in numbers the severity score reported in the graphs.
To improve the significance of the paper I would recommend detailing the quantification method used by the QdB software to give the reader also some insight of how these parameters can be quantified numerically.
Please, also, spell out QdD or add this acronym next to the first time you mention it in the abstract.
Author Response
Thank you for your valuable remarks, please find below our answers.
Answer to Reviewer 1
In the paper “A Proposed Methodology for A Risk Assessment-Based Liposome Development Process” the authors describe an approach to systematically assess the requirement and critical parameters to optimise and tailor the liposomal formulation for a specific application. The quantification of the parameters impact is novel and interesting and is achieved using the Quality by Design (QbD) method whose components are clearly defined in the introduction. The chosen factors of product profile and quality attribute are well listed and justified. The results are also shown and discussed in a clear way and seem legit, even if not so novel.
Comment 1: Anyway, it is not clear how the Risk Assessment (RA) is performed including within the software the previous knowledge, previous experimental experiences and literature background. Moreover, is not specified how the quantification and relative relevance, that represent the main novelty of the paper, are quantified and how the described method is able to quantify in numbers the severity score reported in the graphs.
Answer: Thank you for the remarks. The description of the Risk Assessment process has been extended. Please, find this new part between Line 148-167:
The previous experimental knowledge from the former practical studies is needed to be completed with the information from the relevant literature. This is crucial to a successful RA, as first, the research team has to define the concrete target product (QTPP), then has to select the critical factors, and in the RA process, the interdependence of the critical factors has to be estimated by their severity. The research team members can estimate the risk level of the interactions during the formulation process (process factors, materials, etc.). All the items applied in the RA (the QTPP elements, the CQAs, the CMAs and CPPs) are defined and selected by the research group members and is strongly influenced by their knowledge. Risk is defined as the combination of the probability of occurrence of harm and the severity of that harm. The RA is a systematic process of organising information to support a risk decision to be made within a risk management process. It consists of the identification of hazards and the analysis of risks [20]. Quality risk management tools provide systemic and reproducible methods based on current knowledge about assessing the probability, severity and sometimes detectability of the risk. The methods can be qualitative or quantitative. When the risk is expressed quantitatively, a numerical scale is assigned to the evaluation [20]. The numeric score of the evaluated risks can come from the severity x occurrence (or probability), or sometimes from the severity x occurrence x detectability if the same graded scale is used for the estimation of all. RA software can help in the process, but even in the software supported assessments the selection of risks and the estimations of severity and occurrence are the task and responsibility of the research members, the software helps in making the calculations and giving the visual appearance of the RA results.
Comment 2: To improve the significance of the paper I would recommend detailing the quantification method used by the QdB software to give the reader also some insight of how these parameters can be quantified numerically.
Answer: Thank you for this advice. The Methods section has been improved with a short explanation of the numerical quantification in Line 207-210.
In the software, the qualitative three-level scale, used for the estimation, is linked to a selectable numeric scale (0-10, or 0-100), which gives finally the severity scores of the evaluated risk factors based on mathematical calculations. In this study, the 0-10 scale was used.
Comment 3: Please, also, spell out QdD or add this acronym next to the first time you mention it in the abstract.
Answer: Thank you for noticing this absence, now the acronym has been inserted into the Abstract in Line 18.
Reviewer 2 Report
A very nice and interesting study involving the risk assessment for the liposomes development process via Quality by Design approach was prepared by Németh et al. In general, the manuscript is well-designed, thus I recommend the article to get published after some minor comments.
Comment 1. Language should be checked from a native speaker; some points need further modification so as to be scientifically sound.
Comment 2. Introduction can be elaborated adding a part which will include references about Qbd approach for the development of liposomes . In addition, discussion part should also enlarged, comparing your results with published articles.
Comment 3. At Fig. 3 there are several minor errors i.e. you write “hydrophilic” (is hydrophilic), “hidration” (hydration) etc.
Author Response
Thank you for your valuable remarks, please find below our answers:
Reviewer 2
A very nice and interesting study involving the risk assessment for the liposomes development process via Quality by Design approach was prepared by Németh et al. In general, the manuscript is well-designed, thus I recommend the article to get published after some minor comments.
Comment 1: Language should be checked from a native speaker; some points need further modification so as to be scientifically sound.
Answer: Thank you for the suggestion. By now the whole text has been checked and corrected by a language proofreader.
Comment 2: Introduction can be elaborated adding a part which will include references about Qbd approach for the development of liposomes. In addition, discussion part should also be enlarged, comparing your results with published articles.
Answer: Thank you for the suggestion. This part will give the manuscript a higher quality. The text has been enlarged with the recommended elaboration in Line 128-130, 284-287 and 305-338.
The development process of the liposomes is challenging due to their complex manufacturing processes. The tools of the QbD approach can guide the formulation process to obtain higher-quality liposomal products [18].
Although, the main points of the tables and figures are shreds of evidence from the literature mixed with practical experiences; the systemic collection of all the relevant factors and data in one paper is the novelty of the work. The demonstration of the CMAs and the CPPs parallelly enhances the transparency of their relationships.
Xu. et al. performed a risk analysis study on liposomes prepared using the thin-film hydration technique and loaded with superoxide dismutase via a freeze-thaw cycling technique. They analysed those factors which affect the size, the encapsulation efficiency and the stability of the liposomes. For this evaluation, they checked the properties of the formulation, the process, the analytical method and the instrumentation reliability. They found that the „analytical method” and the „instrument reliability” categories can be well-controlled; therefore their factors are not critical; however, the groups are non-negligible for the selection and settings of the characterisation methods. Their findings, namely the influencing role of the lipid concentration, the cholesterol ratio and the quality of the phospholipids are consistent with our results [31]. Porfire et al. provided a general overview of the QbD approach for liposomes without defining a production process and described methodologies for liposome characterisation as a control strategy in details. Their considerations for selecting QTPPs, CQAs, CMAs and CPPs are reasonable and were built into the tables of this paper. The study emphasised the importance of the determination, investigation and control of the factors that can change the quality, efficacy and safety of the pharmaceutical products. It drew attention to the low number of studies following the regulatory guidelines recommended steps of the QbD [18].
Comment 3: At Fig. 3 there are several minor errors i.e. you write “hydrophilic” (is hydrophilic), “hidration” (hydration) etc.
Answer: Thank you for noticing these errors. A corrected figure (now Figure 4) has been inserted into the manuscript in Line 261.
Reviewer 3 Report
The authors proposed a manuscript whose title is “Proposed Methodology for A Risk Assessment-Based Liposome Development Process”.
The first observation is about the affiliation. Since it is the same for all the authors, it could be better to indicate it only once. Then, I do not know if it is necessary to indicate all the author’s email addresses. Maybe only the corresponding author’s email should be added. However, in case I am wrong, check on the author’s guideline.
Overall comments on the manuscript
English is poor. It should be checked and corrected in all the manuscript.
The preliminary explanation of concepts given in the introduction are not well developed and too much reductive. Authors introduced several concepts in the state of the art section, but they do not explain them deeply, linking to previous works on this Journal or to other cited works.
The organization of the manuscript is not clear. Result section should contain only the output of this study. Table 1 and Table 2 come from considerations obtained from the study of the literature. Also, Figure 1 is mainly the description of the thin layer hydration method. In my opinion, it should not be inserted in the Results section.
After all the evaluation of this manuscript, I regret to say that in my opinion it does not deserve to be published.
I have some comments that in my opinion could improve the manuscript:
Abstract
Line 15. This sentence has no verb. Please check the English: “ The requirements of a liposomal formulation alter depending on the pharmaceutical indication, the target patient population and the corresponding route of administration.”
Line 17. Define the meaning of Material Attributes and Process Parameters, even briefly, in the abstract.
Line 17. Are you sure that “variant” is the correct word in this context?
Line 24. The use of English here is not correct “ the possible factors what have an impact on the quality of the final product and need to be considered”. Please check the syntax, especially on the use of “what” and “need”.
Line 27. QbD. This should be defined in parenthesis at line 21.
An abbreviation list of all the acronyms could be inserted among the Abstract and the Introduction sections.
Introduction and Results sections
Line 33. Morphological definition of liposome is required.
Line 37. “are from this field”. Maybe better “belong to this field”
Line 38. A reference is missing after this sentence. Bibliography about new developed studies should be mentioned.
Line 47. If we are talking about liposomes, that are for definition spherical vesicles, we can only talk about diameter or at least radius.
Line 48. I would say “therefore” in the place of “so”.
Line 51, “The zeta potential value is used to conclude the repulsion or the attraction between the vesicles”. Why you say “conclude”? maybe “define” or “evaluate”?
I would organize the steps of the QbD method in a Table.
The authors say that they made this analysis on the thin layer hydration method for the production of liposomes. However, it is necessary to explain, also qualitatively, how this process is made. Even if this process is well known to the researchers involved in the liposome field.
Table 1. why (distribution), (metabolism) are cited in parenthesis?
Table 2. 0.5 μm < . I would say <0.5
Table 2. unilamellar (in any size) . it is very difficult to have only 1 layer for liposomes of 50-100 micron diameter.
Table 2. between SUV and LUV. Please provide a range diameters for MUV.
I do not agree with the definition of spherical vesicles related to concentric layers. They are always spherical, as definition of liposomes. Concentric layers should be defined as multi-layered or multi-vesicular vesicles.
Table 1 and Table 2. The methods of production have not been considered. Thin layer hydration is one, but there are several more with atmospheric pressure and with high pressure systems.
Table 1 and Table 2, as well as Figure 1, should be inserted elsewhere.
Line 135. This sentence “Liposomes with a charge higher or equal to 10 mV in absolute value are considered as negative or positive” should be inserted in the introduction, where zeta potential is mentioned for the first time. Moreover, liposomes are considered stable with a zeta potential lower than -30 mV or higher than +30 mV.
Line 138. Authors say “PdI values, less or equal to 0.3, are supposed to 138 be the indicator of a monodisperse distribution”. I do not agree with this sentence. A PDI of 0.3 is too large to define a liposome population as monodispersed. In pharmaceutical field, PDI should be less than 0.2 or also less than 0.15 to be considered monodisperse. Pharmaceutic industry prefer a 0.05 PDI, and they also manage to do this.
Author Response
Reviewer
The authors proposed a manuscript whose title is “Proposed Methodology for A Risk Assessment-Based Liposome Development Process”.
Comment 1: The first observation is about the affiliation. Since it is the same for all the authors, it could be better to indicate it only once. Then, I do not know if it is necessary to indicate all the author’s email addresses. Maybe only the corresponding author’s email should be added. However, in case I am wrong, check on the author’s guideline.
Answer: Thank you for these suggestions. The affiliation section has been modified according to the advice and the author’s guideline.
Overall comments on the manuscript
Comment 2: English is poor. It should be checked and corrected in all the manuscript.
Answer: Thank you for the suggestion. By now the whole text has been checked and corrected by a language proofreader.
Comment 3: The preliminary explanation of concepts given in the introduction are not well developed and too much reductive.
Answer: The Introduction section has been completed with definitions, an extended description of the Risk Assessment process and the production method. A figure (Figure 1: Schematic structure of the QbD approach) and a Table (Table 1: Potential methods to prepare liposomes) have been inserted into the section to assist better understanding and to represent more details.
Line 79-81, 99-101, 103-106, 128-130, 148-167, 171-172, 185, 193-199
Comment 4: Authors introduced several concepts in the state of the art section, but they do not explain them deeply, linking to previous works on this Journal or to other cited works.
Answer: Thank you for the comment. This sentence in question was modified, and the concrete literature was nominated, please find in Line 126.
QbD is a systemized, structured, knowledge-, and risk assessment-focused approach, and the potentials of its extension have been shown by Csóka et al. previously [14].
[14] Csóka I, Pallagi E, Paál TL. Extension of quality-by-design concept to the early development phase of pharmaceutical R&D processes. Drug Discov Today 2018. https://doi.org/10.1016/j.drudis.2018.03.012.
Comment 5: The organisation of the manuscript is not clear. Result section should contain only the output of this study. Table 1 and Table 2 come from considerations obtained from the study of the literature. Also, Figure 1 is mainly the description of the thin layer hydration method. In my opinion, it should not be inserted in the Results section.
Answer: The manuscript describes a QbD-based liposome design. In order to help researchers in their future work, the necessary information for design and planning was collected in a systemised and structured way. The main points are from the literature; however, the novel concept in the manuscript, as a result, is the collection of all the relevant factors and data in one paper. The figure mentioned above (now Figure 2) is also a new concept for the demonstrations of the CMAs and the CPPs parallelly to enhance the transparency of their relationships. The figure synthesises and structures the shreds of evidence in the literature and the knowledge from practical experiences.
I have some comments that in my opinion, could improve the manuscript:
Abstract
Comment 6: Line 15. This sentence has no verb. Please check the English: “ The requirements of a liposomal formulation alter depending on the pharmaceutical indication, the target patient population and the corresponding route of administration.”
Answer: Thank you for the remark. This sentence has been corrected; please find in Line 11.
The requirements of a liposomal formulation vary depending on the pharmaceutical indication, the target patient population and the corresponding route of administration.
Comment 7: Line 17. Define the meaning of Material Attributes and Process Parameters, even briefly, in the abstract.
Answer: Thank you for this suggestion. The following definitions have been added to the abstract in Line13-14:
Material Attributes (MAs) (properties and characteristics of the components);
Process Parameters (PPs) (settings of the preparation method)
Comment 8: Line 17. Are you sure that “variant” is the correct word in this context?
Answer: Thank you for the notification. The sentence has been modified; please find in Line 13-14.
Different preparation methods require various Material Attributes (MAs) (properties and characteristics of the components) and Process Parameters (PPs) (settings of the preparation method).
Comment 9: Line 24. The use of English here is not correct “ the possible factors what have an impact on the quality of the final product and need to be considered”. Please check the syntax, especially on the use of “what” and “need”.
Answer: Thank you for the notification. The sentence has been modified; please find in Line 22.
In this paper, the requirements of a liposome formulation prepared via the thin-film hydration preparation technique are presented; furthermore, the possible factors that have an impact on the quality of the final product and have to be considered and specified during the development of a liposomal formulation are identified and collected.
Comment 10: Line 27. QbD. This should be defined in parenthesis at line 21.
Answer: Thank you for noticing this absence, now the acronym has been inserted into the Abstract in Line 18.
Comment 11: An abbreviation list of all the acronyms could be inserted among the Abstract and the Introduction sections.
Answer: Thank you for this quality improving remark. An abbreviation list has been inserted into the suggested place between Line 29-76.
Introduction and Results sections
Comment 12: Line 33. Morphological definition of liposome is required.
Answer: Thank you for this valuable remark. The definition of liposomes made by the European Medicine Agency has been inserted into the text in Line 79-81.
Liposomes are described as artificially prepared vesicles composed of one or more concentric lipid bilayers that are enclosing one or more aqueous compartments by the European Medicine Agency [1].
Comment 13: Line 37. “are from this field”. Maybe better “belong to this field.”
Answer: Thank you for the comment. It has been corrected as your suggestion; please find in Line 85.
Their pharmaceutical existence is essential in the field of cancer therapy, as many of the already marketed liposomal drugs are from belong to this field, and several new studies are in progress in the above-mentioned and newly targeted medical areas as well [2–4].
Comment 14: Line 38. A reference is missing after this sentence. Bibliography about new developed studies should be mentioned.
Answer: Thank you for the notice and the suggestion. The missing reference has been given, and new pieces of literature have been inserted into Line 91.
However, the regulatory authorities need to meet several challenges from the quality, safety and efficacy aspects of the liposome-based products [6] [7].
[6] Hafner A, Lovrić J, Lakoš GP, Pepić I. Nanotherapeutics in the EU: an overview on current state and future directions. Int J Nanomedicine 2014;9:1005–23. https://doi.org/10.2147/IJN.S55359.
[7] Sainz V, Conniot J, Matos AI, Peres C, Zupančič E, Moura L, et al. Regulatory aspects on nanomedicines. Biochem Biophys Res Commun 2015;468:504–10. https://doi.org/10.1016/j.bbrc.2015.08.023.
Comment 15: Line 47. If we are talking about liposomes, that are for definition spherical vesicles, we can only talk about diameter or at least radius.
Answer: Thank you for the remark. Now the sentence contains the diameter world, please find in Line 95.
Based on their definitions, nanoparticles are those nano-objects which have all of their external dimensions diameters in the nanoscale, and there is no significant difference between the lengths of the longest and shortest axes of the particle [8].
Comment 16: Line 48. I would say “therefore” in the place of “so”.
Answer: Thank you for this comment. The sentence has been corrected. Please, find in Line 96.
Therefore, the size of the liposomes and its homogeneity (size distribution) are fundamental features of the systems.
Comment 17: Line 51, “The zeta potential value is used to conclude the repulsion or the attraction between the vesicles”. Why you say “conclude”? maybe “define” or “evaluate”?
Answer: Thank you for the remark. Now the sentence contains the define verb, please find in Line 102.
The zeta potential value is used to define the repulsion or the attraction between the vesicles, and in this way, to predict the stability of the liposome system [10].
Comment 18: I would organise the steps of the QbD method in a Table.
Answer: The steps of the QbD has been organised in Figure 1: Schematic structure of the QbD approach. It is inserted in Line 185.
Reviewer 3
The authors proposed a manuscript whose title is “Proposed Methodology for A Risk Assessment-Based Liposome Development Process”.
Comment 1: The first observation is about the affiliation. Since it is the same for all the authors, it could be better to indicate it only once. Then, I do not know if it is necessary to indicate all the author’s email addresses. Maybe only the corresponding author’s email should be added. However, in case I am wrong, check on the author’s guideline.
Answer: Thank you for these suggestions. The affiliation section has been modified according to the advice and the author’s guideline.
Overall comments on the manuscript
Comment 2: English is poor. It should be checked and corrected in all the manuscript.
Answer: Thank you for the suggestion. By now the whole text has been checked and corrected by a language proofreader.
Comment 3: The preliminary explanation of concepts given in the introduction are not well developed and too much reductive.
Answer: The Introduction section has been completed with definitions, an extended description of the Risk Assessment process and the production method. A figure (Figure 1: Schematic structure of the QbD approach) and a Table (Table 1: Potential methods to prepare liposomes) have been inserted into the section to assist better understanding and to represent more details.
Line 79-81, 99-101, 103-106, 128-130, 148-167, 171-172, 185, 193-199
Comment 4: Authors introduced several concepts in the state of the art section, but they do not explain them deeply, linking to previous works on this Journal or to other cited works.
Answer: Thank you for the comment. This sentence in question was modified, and the concrete literature was nominated, please find in Line 126.
QbD is a systemised, structured, knowledge-, and risk assessment-focused approach, and the potentials of its extension have been shown by Csóka et al. previously [14].
[14] Csóka I, Pallagi E, Paál TL. Extension of quality-by-design concept to the early development phase of pharmaceutical R&D processes. Drug Discov Today 2018. https://doi.org/10.1016/j.drudis.2018.03.012.
Comment 5: The organisation of the manuscript is not clear. Result section should contain only the output of this study. Table 1 and Table 2 come from considerations obtained from the study of the literature. Also, Figure 1 is mainly the description of the thin layer hydration method. In my opinion, it should not be inserted in the Results section.
Answer: The manuscript describes a QbD-based liposome design. In order to help researchers in their future work, the necessary information for design and planning was collected in a systemised and structured way. The main points are from the literature; however, the novel concept in the manuscript, as a result, is the collection of all the relevant factors and data in one paper. The figure mentioned above (now Figure 2) is also a new concept for the demonstrations of the CMAs and the CPPs parallelly to enhance the transparency of their relationships. The figure synthesises and structures the shreds of evidence in the literature and the knowledge from practical experiences.
I have some comments that in my opinion, could improve the manuscript:
Abstract
Comment 6: Line 15. This sentence has no verb. Please check the English: “ The requirements of a liposomal formulation alter depending on the pharmaceutical indication, the target patient population and the corresponding route of administration.”
Answer: Thank you for the remark. This sentence has been corrected; please find in Line 11.
The requirements of a liposomal formulation vary depending on the pharmaceutical indication, the target patient population and the corresponding route of administration.
Comment 7: Line 17. Define the meaning of Material Attributes and Process Parameters, even briefly, in the abstract.
Answer: Thank you for this suggestion. The following definitions have been added to the abstract in Line13-14:
Material Attributes (MAs) (properties and characteristics of the components);
Process Parameters (PPs) (settings of the preparation method)
Comment 8: Line 17. Are you sure that “variant” is the correct word in this context?
Answer: Thank you for the notification. The sentence has been modified; please find in Line 13-14.
Different preparation methods require various Material Attributes (MAs) (properties and characteristics of the components) and Process Parameters (PPs) (settings of the preparation method).
Comment 9: Line 24. The use of English here is not correct “ the possible factors what have an impact on the quality of the final product and need to be considered”. Please check the syntax, especially on the use of “what” and “need”.
Answer: Thank you for the notification. The sentence has been modified; please find in Line 22.
In this paper, the requirements of a liposome formulation prepared via the thin-film hydration preparation technique are presented; furthermore, the possible factors that have an impact on the quality of the final product and have to be considered and specified during the development of a liposomal formulation are identified and collected.
Comment 10: Line 27. QbD. This should be defined in parenthesis at line 21.
Answer: Thank you for noticing this absence, now the acronym has been inserted into the Abstract in Line 18.
Comment 11: An abbreviation list of all the acronyms could be inserted among the Abstract and the Introduction sections.
Answer: Thank you for this quality improving remark. An abbreviation list has been inserted into the suggested place between Line 29-76.
Introduction and Results sections
Comment 12: Line 33. Morphological definition of liposome is required.
Answer: Thank you for this valuable remark. The definition of liposomes made by the European Medicine Agency has been inserted into the text in Line 79-81.
Liposomes are described as artificially prepared vesicles composed of one or more concentric lipid bilayers that are enclosing one or more aqueous compartments by the European Medicine Agency [1].
Comment 13: Line 37. “are from this field”. Maybe better “belong to this field.”
Answer: Thank you for the comment. It has been corrected as your suggestion; please find in Line 85.
Their pharmaceutical existence is essential in the field of cancer therapy, as many of the already marketed liposomal drugs are from belong to this field, and several new studies are in progress in the above-mentioned and newly targeted medical areas as well [2–4].
Comment 14: Line 38. A reference is missing after this sentence. Bibliography about new developed studies should be mentioned.
Answer: Thank you for the notice and the suggestion. The missing reference has been given, and new pieces of literature have been inserted into Line 91.
However, the regulatory authorities need to meet several challenges from the quality, safety and efficacy aspects of the liposome-based products [6] [7].
[6] Hafner A, Lovrić J, Lakoš GP, Pepić I. Nanotherapeutics in the EU: an overview on current state and future directions. Int J Nanomedicine 2014;9:1005–23. https://doi.org/10.2147/IJN.S55359.
[7] Sainz V, Conniot J, Matos AI, Peres C, Zupančič E, Moura L, et al. Regulatory aspects on nanomedicines. Biochem Biophys Res Commun 2015;468:504–10. https://doi.org/10.1016/j.bbrc.2015.08.023.
Comment 15: Line 47. If we are talking about liposomes, that are for definition spherical vesicles, we can only talk about diameter or at least radius.
Answer: Thank you for the remark. Now the sentence contains the diameter world, please find in Line 95.
Based on their definitions, nanoparticles are those nano-objects which have all of their external dimensions diameters in the nanoscale, and there is no significant difference between the lengths of the longest and shortest axes of the particle [8].
Comment 16: Line 48. I would say “therefore” in the place of “so”.
Answer: Thank you for this comment. The sentence has been corrected. Please, find in Line 96.
Therefore, the size of the liposomes and its homogeneity (size distribution) are fundamental features of the systems.
Comment 17: Line 51, “The zeta potential value is used to conclude the repulsion or the attraction between the vesicles”. Why you say “conclude”? maybe “define” or “evaluate”?
Answer: Thank you for the remark. Now the sentence contains the define verb, please find in Line 102.
The zeta potential value is used to define the repulsion or the attraction between the vesicles, and in this way, to predict the stability of the liposome system [10].
Comment 18: I would organise the steps of the QbD method in a Table.
Answer: The steps of the QbD has been organised in Figure 1: Schematic structure of the QbD approach. It is inserted in Line 185. (and attached here as a separate file)
Comment 19: The authors say that they made this analysis on the thin layer hydration method for the production of liposomes. However, it is necessary to explain, also qualitatively, how this process is made. Even if this process is well known to the researchers involved in the liposome field.
Answer: Thank you for this suggestion. A short description of the process has been inserted into the Introduction in Line 193-198.
This method was the first time described and used to prepare the first liposomes by Alec Douglas Bangham and his colleagues in 1965 [22]. Several modified versions of the original technique are existing; however, the basic steps of the process are mutual: 1, preparation of the lipid film from phospholipids and cholesterol, 2, hydration of the thin film with a hydration media, and 3, modification of the numbers of layers and the size of vesicles [23]. The process is described in more details in Figure 2.
Comment 20: Table 1. why (distribution), (metabolism) are cited in parenthesis?
Answer: Thank you for noticing these mistakes. The parentheses have been removed. By now this table has been changed to Table 2 in Line 220.
Comment 21: Table 2. 0.5 μm < . I would say <0.5
Answer: Thank you for the comment. The row has been modified based on the suggestion. By now this table has been changed to Table 3 in Line 227.
Comment 22: Table 2. unilamellar (in any size) . it is very difficult to have only 1 layer for liposomes of 50-100 micron diameter.
Answer: Thank you for the comment. The content of the box has been modified according to the suggestion. By now this table has been changed to Table 3 in Line 227.
Comment 23: Table 2. between SUV and LUV. Please provide a range diameters for MUV.
Answer: Thank you for noticing the absence. The required size (> 100 nm) has been provided. By now this table has been changed to Table 3 in Line 227.
Comment 24: I do not agree with the definition of spherical vesicles related to concentric layers. They are always spherical, as definition of liposomes. Concentric layers should be defined as multi-layered or multi-vesicular.
Answer: Thank you for your valuable remark. The table (now Table 3) has been modified based on the comment.
morphology of liposomes |
spherical vesicles |
self-organised structure |
concentric layers |
multi-layered or multi-vesicular vesicles (MVV) |
Comment 25: Table 1 and Table 2. The methods of production have not been considered. Thin layer hydration is one, but there are several more with atmospheric pressure and with high pressure systems.
Answer: Thank you for this golden remark. The text has been improved with a new table (Table 1, see below) in 199 and some referring sentences in Line 237-240.
The newly inserted table is the following:
Preparation methods |
Subtypes |
Comments |
mechanical dispersion methods |
sonication |
- the CPPs vary based on the selected preparation method; therefore the definition of the production technique has to be the first step of every liposome formulation process - the properties of the liposomes (e.g. number of lamellas, size and distribution of vesicles) |
French pressure cell - extrusion |
||
freeze-thawed liposomes |
||
lipid film hydration techniques |
||
hydration of proliposomes |
||
micro emulsification, |
||
membrane extrusion |
||
dual asymmetric centrifugation |
||
heating method |
||
solvent dispersion methods |
ether injection |
|
ethanol injection |
||
reverse-phase evaporation |
||
solvent spherule method |
||
detergent removal methods |
dialysis |
|
detergent removal of mixed micelles |
||
gel-permeation chromatography |
||
novel approaches |
electroformation methods |
|
microfluidic methods |
||
supercritical fluids method |
||
freeze-drying of double emulsions |
||
membrane contractor method |
||
curvature-tuning |
||
biometric reaction for vesicular self-assembly |
||
Mozafari method |
The preparation method (Table 1) define the CPPs of the liposome formulation process; therefore, a production technique that provides the aimed CQAs has to be chosen before the investigation of the CMAs and CPPs. In this paper, the thin-film hydration method-related factors are presented.
Comment 26: Table 1 and Table 2, as well as Figure 1, should be inserted elsewhere.
Answer: Although, the main points of the tables and figures are from the literature; the collection of all the relevant factors and data in one paper is the novelty of the manuscript, and therefore, need to be placed in the Results section. Please accept this proposal. The demonstration of the CMAs and the CPPs parallelly is a new concept to enhance the transparency of their relationships.
Comment 27: Line 135. This sentence “Liposomes with a charge higher or equal to 10 mV in absolute value are considered as negative or positive” should be inserted in the introduction, where zeta potential is mentioned for the first time. Moreover, liposomes are considered stable with a zeta potential lower than -30 mV or higher than +30 mV.
Answer: Thank you for this suggestion. The sentence now has been inserted into the Introduction section in Line 103-106.
Comment 28: Line 138. Authors say “PdI values, less or equal to 0.3, are supposed to 138 be the indicator of a monodisperse distribution”. I do not agree with this sentence. A PDI of 0.3 is too large to define a liposome population as monodispersed. In pharmaceutical field, PDI should be less than 0.2 or also less than 0.15 to be considered monodisperse. Pharmaceutic industry prefer a 0.05 PDI, and they also manage to do this.
Answer: The comment is right. Thank you for noticing the drafting error. The sentence has been corrected, enlarged, and parallelly inserted into the Introduction in Line 99-101.
PdI values less or equal to 0.3 are supposed to be the indicator of distribution with acceptably low polydispersity. In the case of lipid-based nanocarriers, formulations with a PdI of 0.3 and bellow is acceptable and the indicator of a homogenous population of the vesicles [9].
Comment 19: The authors say that they made this analysis on the thin layer hydration method for the production of liposomes. However, it is necessary to explain, also qualitatively, how this process is made. Even if this process is well known to the researchers involved in the liposome field.
Answer: Thank you for this suggestion. A short description of the process has been inserted into the Introduction in Line 193-198.
This method was the first time described and used to prepare the first liposomes by Alec Douglas Bangham and his colleagues in 1965 [22]. Several modified versions of the original technique are existing; however, the basic steps of the process are mutual: 1, preparation of the lipid film from phospholipids and cholesterol, 2, hydration of the thin film with a hydration media, and 3, modification of the numbers of layers and the size of vesicles [23]. The process is described in more details in Figure 2.
Comment 20: Table 1. why (distribution), (metabolism) are cited in parenthesis?
Answer: Thank you for noticing these mistakes. The parentheses have been removed. By now this table has been changed to Table 2 in Line 220.
Comment 21: Table 2. 0.5 μm < . I would say <0.5
Answer: Thank you for the comment. The row has been modified based on the suggestion. By now this table has been changed to Table 3 in Line 227.
Comment 22: Table 2. unilamellar (in any size) . it is very difficult to have only 1 layer for liposomes of 50-100 micron diameter.
Answer: Thank you for the comment. The content of the box has been modified according to the suggestion. By now this table has been changed to Table 3 in Line 227.
Comment 23: Table 2. between SUV and LUV. Please provide a range diameters for MUV.
Answer: Thank you for noticing the absence. The required size (> 100 nm) has been provided. By now this table has been changed to Table 3 in Line 227.
Comment 24: I do not agree with the definition of spherical vesicles related to concentric layers. They are always spherical, as definition of liposomes. Concentric layers should be defined as multi-layered or multi-vesicular.
Answer: Thank you for your valuable remark. The table (now Table 3) has been modified based on the comment.
morphology of liposomes |
spherical vesicles |
self-organised structure |
concentric layers |
multi-layered or multi-vesicular vesicles (MVV) |
Comment 25: Table 1 and Table 2. The methods of production have not been considered. Thin layer hydration is one, but there are several more with atmospheric pressure and with high pressure systems.
Answer: Thank you for this golden remark. The text has been improved with a new table (Table 1, see below) in 199 and some referring sentences in Line 237-240.
The newly inserted table is the following:
Preparation methods |
Subtypes |
Comments |
mechanical dispersion methods |
sonication |
- the CPPs vary based on the selected preparation method; therefore the definition of the production technique has to be the first step of every liposome formulation process - the properties of the liposomes (e.g. number of lamellas, size and distribution of vesicles) |
French pressure cell - extrusion |
||
freeze-thawed liposomes |
||
lipid film hydration techniques |
||
hydration of proliposomes |
||
micro emulsification, |
||
membrane extrusion |
||
dual asymmetric centrifugation |
||
heating method |
||
solvent dispersion methods |
ether injection |
|
ethanol injection |
||
reverse-phase evaporation |
||
solvent spherule method |
||
detergent removal methods |
dialysis |
|
detergent removal of mixed micelles |
||
gel-permeation chromatography |
||
novel approaches |
electroformation methods |
|
microfluidic methods |
||
supercritical fluids method |
||
freeze-drying of double emulsions |
||
membrane contractor method |
||
curvature-tuning |
||
biometric reaction for vesicular self-assembly |
||
Mozafari method |
The preparation method (Table 1) define the CPPs of the liposome formulation process; therefore, a production technique that provides the aimed CQAs has to be chosen before the investigation of the CMAs and CPPs. In this paper, the thin-film hydration method-related factors are presented.
Comment 26: Table 1 and Table 2, as well as Figure 1, should be inserted elsewhere.
Answer: Although, the main points of the tables and figures are from the literature; the collection of all the relevant factors and data in one paper is the novelty of the manuscript, and therefore, need to be placed in the Results section. Please accept this proposal. The demonstration of the CMAs and the CPPs parallelly is a new concept to enhance the transparency of their relationships.
Comment 27: Line 135. This sentence “Liposomes with a charge higher or equal to 10 mV in absolute value are considered as negative or positive” should be inserted in the introduction, where zeta potential is mentioned for the first time. Moreover, liposomes are considered stable with a zeta potential lower than -30 mV or higher than +30 mV.
Answer: Thank you for this suggestion. The sentence now has been inserted into the Introduction section in Line 103-106.
Comment 28: Line 138. Authors say “PdI values, less or equal to 0.3, are supposed to 138 be the indicator of a monodisperse distribution”. I do not agree with this sentence. A PDI of 0.3 is too large to define a liposome population as monodispersed. In pharmaceutical field, PDI should be less than 0.2 or also less than 0.15 to be considered monodisperse. Pharmaceutic industry prefer a 0.05 PDI, and they also manage to do this.
Answer: The comment is right. Thank you for noticing the drafting error. The sentence has been corrected, enlarged, and parallelly inserted into the Introduction in Line 99-101.
PdI values less or equal to 0.3 are supposed to be the indicator of distribution with acceptably low polydispersity. In the case of lipid-based nanocarriers, formulations with a PdI of 0.3 and bellow is acceptable and the indicator of a homogenous population of the vesicles [9].

Round 2
Reviewer 3 Report
Authors provided a revised version of the paper that I have previously revised. Even if the manuscript is improved, I still think that English should be checked in the new written parts.
Moreover, the bibliography is still poor and needs to be enlarged.
I have some other issues:
Line 79. “bellow”. Must be “below”.
Line 82. “charge” should be “average surface charge”
As said in the first paragraph, I suggest to check again the English in the new paragraph among Line 115 and Line 130
Define CPP in Table 1 captions, maybe
Table 1. Methods should be accompanied by references of each technique.
I would say that among novel processes, only the high pressure processes can be annoverated.
Line 152. Figure 2 is named in this Line, but appears much after.
Line 159. I would say “at the end” not “finally”, which has a different meaning.
Table 3. what do you mean with conventional liposomes? Generally it is a way to say that liposomes have been produced with conventional techniques. However, there are conventional techniques used to produce immune, cationic and stimuli responsibve liposome.clarify this aspect.
Table 3. multivescicular vesicles. They are not necessary concentric. They can include also disjointed layers into the inner core of the biggest layer.
Author Response
Dear Reviewer 3,
We would like to thank your work in assessing our manuscript. It is highly appreciated and welcomed. As a result of following the useful advice found in the reviewer report, we have improved the manuscript. The texts of the Introduction part and the bibliography have been extended; moreover, Table 3 has been completed in correspondence to the requests. The grammar of the text has been revised.
You will find below the issues addressed and the ways how they have been answered.
Answers to the review report:
Comment 1: Authors provided a revised version of the paper that I have previously revised. Even if the manuscript is improved, I still think that English should be checked in the new written parts.
Moreover, the bibliography is still poor and needs to be enlarged.
Answer: Thank you for the suggestion. The Introduction part of the manuscript has been further improved and completed with more bibliography. The grammar of the text has been revised again. The list of new references is the following:
[2] Sercombe L, Veerati T, Moheimani F, Wu SY, Sood AK, Hua S. Advances and challenges of liposome assisted drug delivery. Front Pharmacol 2015;6:1–13. https://doi.org/10.3389/fphar.2015.00286.
[4] Cattel L, Ceruti M, Dosio F. From conventional to stealth liposomes a new frontier in cancer chemotherapy. Tumori 2003;89:237–49. https://doi.org/10.1177/030089160308900302.
[5] Riaz MK, Riaz MA, Zhang X, Lin C, Wong KH, Chen X, et al. Surface functionalization and targeting strategies of liposomes in solid tumor therapy: A review. Int J Mol Sci 2018;19. https://doi.org/10.3390/ijms19010195.
[6] Tsermentseli SK, Kontogiannopoulos KN, Papageorgiou VP, Assimopoulou AN. Comparative study of pEgylated and conventional liposomes as carriers for shikonin. Fluids 2018;3:1–16. https://doi.org/10.3390/fluids3020036.
[7] Madni MA, Sarfraz M, Rehman M, Ahmad M, Akhtar N, Ahmad S, et al. Liposomal drug delivery: A versatile platform for challenging clinical applications. J Pharm Pharm Sci 2014;17:401–26. https://doi.org/10.18433/j3cp55.
[8] Hansen AH, Mouritsen OG, Arouri A. Enzymatic action of phospholipase A2 on liposomal drug delivery systems. Int J Pharm 2015;491:49–57. https://doi.org/10.1016/j.ijpharm.2015.06.005.
[9] Samoshin V V. Fliposomes: Stimuli-triggered conformational flip of novel amphiphiles causes an instant cargo release from liposomes. Biomol Concepts 2014;5:131–41. https://doi.org/10.1515/bmc-2014-0002.
[10] Perez-Soler R. Liposomes as carriers of anticancer agents. Drug News Perspect 1990;3:287–91.
[11] Daraee H, Etemadi A, Kouhi M, Alimirzalu S, Akbarzadeh A. Application of liposomes in medicine and drug delivery. Artif Cells, Nanomedicine Biotechnol 2016;44:381–91. https://doi.org/10.3109/21691401.2014.953633.
[23] Yu LX. Pharmaceutical quality by design: Product and process development, understanding, and control. Pharm Res 2008;25:781–91. https://doi.org/10.1007/s11095-007-9511-1.
[28] Sipos B, Szabó-Révész P, Csóka I, Pallagi E, Dobó DG, Bélteky P, et al. Quality by design based formulation study of meloxicam-loaded polymeric micelles for intranasal administration. Pharmaceutics 2020;12:1–29. https://doi.org/10.3390/pharmaceutics12080697.
[29] Katona G, Balogh GT, Dargó G, Gáspár R, Márki Á, Ducza E, et al. Development of meloxicam-human serum albumin nanoparticles for nose-to-brain delivery via application of a quality by design approach. Pharmaceutics 2020;12. https://doi.org/10.3390/pharmaceutics12020097.
[30] Mukhtar M, Pallagi E, Csóka I, Benke E, Farkas Á, Zeeshan M, et al. Aerodynamic properties and in silico deposition of isoniazid loaded chitosan/thiolated chitosan and hyaluronic acid hybrid nanoplex DPIs as a potential TB treatment. Int J Biol Macromol 2020;165:3007–19. https://doi.org/10.1016/j.ijbiomac.2020.10.192.
[38] Patil YP, Jadhav S. Novel methods for liposome preparation. Chem Phys Lipids 2014;177:8–18. https://doi.org/10.1016/j.chemphyslip.2013.10.011.
[39] Trucillo P, Campardelli R, Reverchon E. Liposomes: From bangham to supercritical fluids. Processes 2020;8:1–15. https://doi.org/10.3390/pr8091022.
[40] Maja L, Željko K, Mateja P. Sustainable technologies for liposome preparation. J Supercrit Fluids 2020;165. https://doi.org/10.1016/j.supflu.2020.104984.
[47] Elgharbawy H, Morsy R. Preparation and Physicochemical Evaluation of Magnetoliposomes as Drug Carriers for 5-Fluorouracile Preparation and Physicochemical Evaluation of Magnetoliposomes as Drug Carriers for 5-Fluorouracile 2016.
[48] Li N, Shi A, Wang Q, Zhang G. Multivesicular liposomes for the sustained release of Angiotensin I-Converting Enzyme (ACE) inhibitory peptides from peanuts: Design, characterization, and in vitro evaluation. Molecules 2019;24. https://doi.org/10.3390/molecules24091746.
Comment 2: Line 79. “bellow”. Must be “below”.
Answer: Thank you for this notification. It has been corrected; please find now in Line 96.
Comment 3: Line 82. “charge” should be “average surface charge”
Answer: Thank you for this suggestion. It has been modified; please find now in Line 99.
Comment 4: As said in the first paragraph, I suggest to check again the English in the new paragraph among Line 115 and Line 130
Answer: Thank you for this helpful advice. The mentioned part has been modified; please find now in Line 132-152.
Comment 5: Define CPP in Table 1 captions, maybe
Answer: Thank you for this suggestion. We decided to write out the acronym in the text of the table.
Comment 6: Table 1. Methods should be accompanied by references of each technique.
Answer: References for the methods mentioned in Table 1 are added in Line 194-203.
Comment 7: I would say that among novel processes, only the high pressure processes can be annoverated.
Answer: Thank you for this comment. The content of Table 1 has been revised.
Comment 8: Line 152. Figure 2 is named in this Line, but appears much after.
Answer: Yes, it is true. Figure 2 is also mentioned in Line 205, and the figure is inserted in Line 209; therefore, we decided to remove the cited sentence from Line 152 (now Line 172).
Comment 9: Line 159. I would say “at the end” not “finally”, which has a different meaning.
Answer: Thank you for this suggestion. It has been modified; please find now in Line 179.
Comment 10: Table 3. what do you mean with conventional liposomes? Generally, it is a way to say that liposomes have been produced with conventional techniques. However, there are conventional techniques used to produce immune, cationic and stimuli responsible liposome. Clarify this aspect.
Answer: Although the grouping of liposomes based on their preparation techniques exists, we would like to refer to the first generation of these vesicles as conventional liposomes. Conventional liposomes are those that made up of neutral and/or negatively charged phospholipids and cholesterol without any modification and thus have short circulation time in the blood system.
[4] Cattel L, Ceruti M, Dosio F. From conventional to stealth liposomes a new frontier in cancer chemotherapy. Tumori 2003;89:237–49. https://doi.org/10.1177/030089160308900302.
The Introduction part of the manuscript has been completed with a part referring to the liposomal generation; please find in Line 63-78.
Comment 11: Table 3. multivesicular vesicles. They are not necessary concentric. They can include also disjointed layers into the inner core of the biggest layer.
Answer: Thank you for this useful comment. Table 3 has been modified according to this information.
morphology of liposomes |
spherical vesicles |
self-organised structure |
concentric layers |
multi-layered vesicles |
|
spherical with multiple non‑concentric lipid vesicles inside |
multivesicular liposome (MVL) |
[48] Li N, Shi A, Wang Q, Zhang G. Multivesicular liposomes for the sustained release of Angiotensin I-Converting Enzyme (ACE) inhibitory peptides from peanuts: Design, characterization, and in vitro evaluation. Molecules 2019;24. https://doi.org/10.3390/molecules24091746.
Yours sincerely,
the authors of the manuscript
Round 3
Reviewer 3 Report
Paper has much improved, if we compare this version to the previous ones.
I was really impressed from the energy and effort that Authors put in this paper.
English was improved, also after a second revision of the manuscript.
I have only a couple of minor issues, after which I think that the paper can be surely published. Authors deserve it.
Line 131. Reevaluated? What do you mean ? maybe “evaluated/measured again”?
Figure 2. Don’t you have a high definition quality of this figure?
Author Response
Dear Reviewer,
Comments and remarks obtained from the Reviewers helped a lot in the improvement process of the paper. The advice showed the way how to overcome the weaknesses of the manuscript, and therefore we are grateful for them.
Comment 1: Line 131. Reevaluated? What do you mean? maybe “evaluated/measured again”?
Answer: We meant that the result of the first Risk Assessment usually needs to be modified, evaluated again or differently when more experiences are obtained in the field, mostly from the practical experiences and results.
The expression has been changed according to the comment in Line 131.
Comment 2: Figure 2. Don’t you have a high definition quality of this figure?
Answer: We tried to convert the figure into a better quality file. We hope that we have succeeded.
The kind work and the valuable advice of the Reviewer are highly appreciated.
Yours sincerely,
the authors
